The impact of the route of administration on the efficacy and safety of the drug therapy for patent ductus arteriosus in premature infants: a systematic review and meta-analysis

Luo Hanwen
He Jianghua
Xu Xiaoming
Chen Hongju cilla_8597@163.com
Shi Jing shijing@scu.edu.cn
Department of Pediatrics, West China Second University Hospital/ Key Laboratory of Birth Defects and Related Diseases of Women and Children Ministry of Education, Sichuan University , Chengdu , Sichuan , China
Yan Yuanliang
Electronic publication date: 2024 Jan 29
Publication date: 2024
Volume: 12
Electronic Location ID: e16591
Received 2023 Aug 31; Accepted 2023 Nov 14
Copyright: ©2024 Luo et al.
Copyright year: 2024
Copyright holder: Luo et al.
License: This is an open access article distributed under the terms of the Creative Commons Attribution License, which permits unrestricted use, distribution, reproduction and adaptation in any medium and for any purpose provided that it is properly attributed. For attribution, the original author(s), title, publication source (PeerJ) and either DOI or URL of the article must be cited.
License URL: https://creativecommons.org/licenses/by/4.0/

Keywords: Patent ductus arteriosus (PDA), The premature, Drug therapy, Adverse events, Administration

Funding: The authors received no funding for this work.

==============================
Background

This systematic review and meta-analysis aims to explore the potential impact of the route of administration on the efficacy of therapies and occurrence of adverse events when administering medications to premature infants with patent ductus arteriosus (PDA).

Method

The protocol for this review has been registered with PROSPERO (CRD 42022324598). We searched relevant studies in PubMed, Embase, Cochrane, and the Web of Science databases from March 26, 1996, to January 31, 2022.

Results

A total of six randomized controlled trials (RCTs) and five observational studies were included for analysis, involving 630 premature neonates in total. Among these infants, 480 were in the ibuprofen group (oral vs. intravenous routes), 78 in the paracetamol group (oral vs. intravenous routes), and 72 in the ibuprofen group (rectal vs. oral routes). Our meta-analysis revealed a significant difference in the rate of PDA closure between the the initial course of oral ibuprofen and intravenous ibuprofen groups (relative risk (RR) = 1.27, 95% confidence interval (CI) [1.13–1.44]; P < 0.0001, I2 = 0%). In contrast, the meta-analysis of paracetamol administration via oral versus intravenous routes showed no significant difference in PDA closure rates (RR = 0.86, 95% CI [0.38–1.91]; P = 0.71, I2 = 76%). However, there was no statistically significant difference in the risk of adverse events or the need for surgical intervention among various drug administration methods after the complete course of drug therapy.

Conclusion

This meta-analysis evaluated the safety and effectiveness of different medication routes for treating PDA in premature infants. Our analysis results revealed that compared with intravenous administration, oral ibuprofen may offer certain advantages in closing PDA without increasing the risk of adverse events. Conversely, the use of paracetamol demonstrated no significant difference in PDA closure and the risk of adverse events between oral and intravenous administration.

Introduction

The ductus arteriosus plays a crucial role in the circulation of both normal fetuses and those with severe congenital heart disease, as it facilitates shunting between systemic and pulmonary circulations. Under normal prenatal conditions, the ductus arteriosus acts as a bridge in the circulation of newborns, allowing most right ventricular output to bypass pulmonary circulation (Alvarez & McBrien, 2018). In full-term infants, ductus closure typically occurs within 24 to 48 h after birth, making the transformation from fetal to newborn circulation. Various factors influence the timing of functional and subsequent anatomic closure of ductus arteriosus.

Functional closure primarily involves a reduction in prostaglandins and an increase in oxygen levels, mediated by factors such as potassium channels, endothelin-1, and isoprostanes. Following functional closure, anatomic closure entails the formation of an intimal cushion, disassembly of the internal elastic lamina, loss of elastic fiber in the medial layer, smooth muscle cell proliferation and migration, extracellular matrix production, endothelial cell proliferation, and interactions between blood cells (Ovalı, 2020). Once these processes are complete, the ductus arteriosus closes permanently and stably.

However, in premature infants, the ductus may fail to close, resulting in a condition known as patent ductus arteriosus (PDA). The incidence of PDA is far greater in premature infants due to their immaturity compared with full-term babies. The immature ductus is highly sensitive to the vasodilation effect caused by nitric oxide and prostaglandin E2, which hinder functional closure. Moreover, factors such as decreased intrinsic tone, diminished ductal muscle fibers, and reduced subendothelial cushions can contribute to failed ductus closure. Approximately 70% of extremely preterm newborns (born at less than 28 weeks of gestation) require medical or surgical intervention to close the PDA (Alvarez & McBrien, 2018). Babies born at earlier gestational ages are more likely to require such treatments. Failure to close the ductus can result in significant left-to-right shunting, leading to complications such as renal, gastrointestinal, and cerebral effects, including intraventricular hemorrhage (IVH), spontaneous intestinal perforation, necrotizing enterocolitis (NEC), bronchopulmonary dysplasia (BPD), renal dysfunction, and even death. This condition is known as hemodynamically significant PDA (hsPDA). In addition to the clinical manifestations, echocardiography also holds significant importance in diagnosing hsPDA. This diagnostic approach is particularly useful for early detection of a noteworthy left-to-right shunting in cases of PDA, especially in extremely premature babies, where the closure can be significantly delayed (Chiruvolu, Punjwani & Ramaciotti, 2009).

In clinical practice, the primary treatments for patients with hsPDA encompass surgical ligation and pharmacological interventions. First-line medications for hsPDA involve non-selective cyclooxygenase (COX) inhibitors such as indomethacin and ibuprofen. These inhibitors hinder the conversion of arachidonic acid into prostaglandins, thereby inducing vasoconstriction and promoting ductus closure. Recently, a relatively recent addition to the treatment repertoire is paracetamol, which operates differently from indomethacin and ibuprofen. Paracetamol works by inhibiting peroxidase (POX) and enhancing the resistance of pulmonary vessels, devoid of any anti-inflammatory effects.

However, the use of COX and POX inhibitors has raised concerns regarding potential adverse events. A study by Ndour et al. (2020) involving 227 infants with a mean gestational age (GA) of 27 weeks (range 24–33) reported notable adverse outcomes. Among the infants treated with ibuprofen, 12 (5%) experienced intestinal perforation, seven (3%) had necrotizing enterocolitis, 25 (11%) developed acute renal failure, and five (2%) exhibited thrombocytopenia. In a study conducted by Schindler et al. (2021) which focused on paracetamol administration in preterm infants born at 29 weeks of gestation, adverse events included intraventricular hemorrhage (four out of 29, 13.8%), pulmonary hemorrhage (two out of 29, 6.9%), chronic lung disease (12 out of 29, 44.4%), necrotizing enterocolitis (two out of 29, 6.9%), sepsis (seven out of 29, 24.1%), retinopathy of prematurity (13 out of 29, 48.1%), hepatic impairment (one out of 29, 3.4%), and renal impairment (nine out of 29, 31.0%). Similarly, a study by Davidson et al. (2021) unveiled adverse events tied to the use of acetaminophen and indomethacin in treating hemodynamically significant PDAs among VLBW (very low birth weight) infants. These events included BPD (82% for Acetaminophen, 65% for Indomethacin), NEC (12% for Acetaminophen, 15% for Indomethacin), IVH (6% for Acetaminophen, 15% for Indomethacin), ROP (12% for Acetaminophen, 5% for Indomethacin), and sepsis (24% for Acetaminophen, 30% for Indomethacin) (Davidson et al., 2021). Furthermore, other adverse occurrences, such as hyperbilirubinemia, localized bowel perforation, and periventricular leucomalacia (PVL), may potentially arise due to medication usage or as a result of the immaturity and persistent left-to-right shunting through the ductus of hsPDA that remained unsealed. These complications can potentially lead to enduring consequences and incur significant treatment costs. It’s worth noting that ibuprofen, with its high protein binding rate of 99% and substantial concentration, may compete with bilirubin for albumin binding, potentially elevating the risk of hyperbilirubinemia.

Given the potential for these complications, early diagnosis and effective therapy are of significant importance for the management of hsPDA, with the route of administration plays a critical role in ensuring optimal therapeutic outcomes.

Intravenous and oral administration are the main routes of drug administration. The oral route offers a range of advantages, including convenience, safety, and cost-effectiveness However, it has its drawbacks, such as slow absorption, uneven distribution, vulnerability to gastrointestinal contents, influences of gastrointestinal peristalsis, and susceptibility to the first-pass effect. Additionally, certain oral pharmaceutical preparations may possess high osmolality, potentially posing risks to the immature gastrointestinal tract. On the other hand, intravenous administration circumvents the first-pass drug effect, enabling direct entry of drug into the systemic circulation and rapid onset of its effects. Nonetheless, it comes with disadvantages, including higher cost, complexity in execution, and increased safety concerns. It is noteworthy that during the neonatal period, due to variations in gastric emptying times or insufficient gastric acid secretion, the oral absorption of many drugs in neonates becomes unreliable and unpredictable. Consequently, the primary route of administration for neonates often defaults to the intravenous route. Exceptions exist, with only a few drugs, such as caffeine, capable of achieving swift and effective blood drug concentrations.

Currently, there exists a body of clinical reports that explore the use of oral, intravenous, or rectal routes of administration treatments for PDA in premature infants (Lago et al., 2002; Castaldo et al., 2023). For instance, indomethacin primarily finds its use in intravenous administration due to its limited and erratic oral bioavailability. In contrast, ibuprofen and acetaminophen offer flexibility in their administration, as they can be delivered intravenously, orally, or rectally. The oral and rectal routes for ibuprofen and acetaminophen fall under off-label usage. It is imperative to recognize that oral formulations of ibuprofen and acetaminophen possess high osmolality and have been associated with cases of necrotizing enterocolitis in premature infants (Gouyon & Kibleur, 2010). Recent Cochrane systematic analysis has indicated that oral ibuprofen outperforms its intravenous counterpart in terms of closing PDA (as evidenced by five studies involving 406 infants; typical RR: 0.38, 95% CI [0.26–0.56] moderate-quality evidence). However, the Cochrane study featured only a limited number of randomized controlled studies (RCT) with small sample sizes, and not all studies reported all adverse events. Currently, no systematic analysis exists that compares the efficacy and safety of administered orally acetaminophen with other routes of administration.

This study, however, takes a more comprehensive approach. It focused on thoroughly examining the efficacy and safety associated with different routes of administration for PDA treatment among preterm infants. We included a broader range of studies (including both RCTs and observational studies), and conducted a meta-analysis comparing PDA closure rates and the occurrence of adverse events across various routes for the same drug. Our objective is to provide valuable insights and guidance regarding the selection of the most suitable drug administration route for the treatment of PDA in premature infants.

Materials & Methods

This systematic review and meta-analysis was reported according to the Preferred Reporting Items for Systematic Reviews and Meta-Analyses (PRISMA) statement (Page et al., 2021) and the Cochrane Handbook for Systematic Reviews of Intervention (Higgins et al., 2011). The protocol for this review has been registered with PROSPERO (CRD42022324598).

Eligibility criteria

Types of studies

RCTs and non-randomized studies written in English that compared different routes of administration using the same medication for the treatment of PDA among premature infants were included in the meta-analysis.

Participants

Our inclusion criteria specified preterm infants born at less than 37 weeks who received either COX or POX inhibitors for the treatment of PDA within the first 28 postnatal days.

We excluded studies that met any of the following criteria: (1) animal studies; (2) redundant or unavailable data for meta-analysis; (3) literature reviews, commentaries, expert opinions, or clinical guidelines; (5) studies without full-text availability; (6) studies written in languages other than English.

Interventions and comparisons

Interventions including indomethacin, ibuprofen, or acetaminophen were administered to facilitate the closure of the PDA in preterm infants.

Different routes of administration were compared as follows:

(1) oral versus intravenous administration;

(2) rectal versus oral administration;

(3) rectal versus intravenous administration.

Outcome measures

Primary outcomes

The primary outcomes were defined as follows:

(1) PDA closure rate, as confirmed by echocardiographic criteria after the first course of drug therapy;

(2) PDA closure rate, as confirmed by echocardiographic criteria after the total course of drug therapy;

(3) The need for surgical ligation of PDA following the allocated treatment.

Secondary outcomes

The secondary outcome was adverse events (ADE).

ADEs are defined as unfavorable or harmful outcomes that occur during or after the use of a drug or other intervention. However, adverse events may not necessarily be causally linked to the intervention. On the other hand, adverse effects, also referred to as harm, pertain to adverse events where there exists at least a reasonable possibility of a causal relationship between the intervention and the event (Rossini, 2010).

Specific secondary outcomes included:

• Bronchopulmonary dysplasia (BPD)/chronic lung disease (CLD), defined as the need for supplemental oxygen at 36 weeks’ postmenstrual age.

• Severe retinopathy of premature (ROP), requiring laser treatment, or the development of stage III ROP or more severe ROP.

• Necrotizing enterocolitis (NEC)/gastrointestinal pathology (GIP) at any stage.

• Intraventricular hemorrhage at all stages.

• Localized bowel perforation/spontaneous intestinal perforation (SIP).

• Gastrointestinal bleeding (GIB)/gastrointestinal hemorrhage (GIH).

• Sepsis.

• Periventricular leukomalacia (PVL).

• Oliguria/decrease in urine output, defined as urine output <one mL/kg/h during an 8-hour collection.

• Renal failure/acute kidney injury (AKI), defined as creatinine >1.2 mg/dL.

• Pulmonary hemorrhage.

• Plasma creatinine levels before and after treatment.

• Thrombocytopenia.

• Pulmonary hypertension.

• Hyperbilirubinemia.

• Plasma bilirubin before and after treatment.

• Mortality during the course of drug treatment.

Search strategy

We searched PubMed, Embase, the Cochrane Central Register of ControlledTrials (CENTRAL), and the Web of Science from March 26, 1996 to January 31, 2022 using the Medical Subject Headings (MeSH) terms: (“premature infant” [MeSH Terms] OR “preterm infant” [MeSH Terms] OR “neonatal prematurity” OR “low birthweight infant” [MeSH] OR “extremely low birth weight infant” [MeSH] OR “very low birth weight Infant” [MeSH] OR “extremely premature infant” [MeSH] OR “very premature infants” [MeSH]) AND (“ductus arteriosus, patent” [MeSH Terms] OR “patent ductus arteriosus” OR “patency of the ductus arteriosus”) AND (“Drug Therapy” [MeSH Terms] OR“ibuprofen” [MeSH Terms] OR “paracetamol” [MeSH Terms]).The search strategy is presented in Appendix S1.

Data extraction and meta-analysis

Two authors independently extracted data from included studies. Any disagreements were resolved through the discussion with a third reviewer until a consensus was reached. Data analysis was performed using the software RevMan and STATA, with results expressed as relative risk (RR) and 95% confidence intervals (CI). A heterogeneity test was done using the I2 statistic and P value. A random-effects model was employed if significant heterogeneity was detected (P < 0.05 and/or I2 > 50%); otherwise, a fixed-effects model was adopted. Sensitivity analyses were performed and funnel plots were used to assess publication bias.

Quality assessment

Two reviewers independently assessed the risk of bias within individual studies using the Jadad scale and the Newcastle–Ottawa scale (NOS). All discrepancies will be resolved by discussion and consensus. The modified Jadad scale consists of six items: (a) randomization; (b) blinding; (c) description of withdrawals and dropouts; (d) inclusion/exclusion criteria; (e) adverse effects; (f) statistical analysis. Scores on this scale range from 0 to 8 points, with higher scores indicating better study quality. Scores of one to three points signify low quality, while scores of four to eight indicates high quality. The NOS evaluation encompassed the following items: (1) Adequacy of case definition; (2) representativeness of cases; (3) selection of controls; (4) definition of controls; (5) comparability; (6) ascertainment of exposure; (7) uniform method of ascertainment for cases and controls; (8) non-response rate.

Results

Literature search and screening results

The initial search in databases yielded a total of 5,010 articles: PubMed (n = 748), Embase (n = 2, 854), Cochrane (n = 452), and the Web of Science (n = 956). After removing 1,557 duplicates, 3,453 articles were reviewed. Among them, 3,442 articles were excluded based on literature type and content (Fig. 1). Subsequently, the remaining articles underwent a thorough review, focusing on indications and outcomes. Finally, six RCTs (Gokmen et al., 2011; Erdeve et al., 2012; Cherif et al., 2008; Edison et al., 2022; Dabas & Khadgawat, 2014; Demir et al., 2017) and 5 observational studies (Abushanab et al., 2021; Olukman et al., 2012; Gover et al., 2022; Sancak et al., 2016; El-Khuffash et al., 2014) were included in this meta-analysis. Characteristics of these studies are summarized in Table 1.

Figure 1 Flow chart.

Table 1 Information of included studies.

First author	Year	Country	Study design	Sample size
(treatment A/ treatment B)	Follow-up	Treat ment A	Treatment B	
						Route	Dosage	Gestational age	Birth weight
(g)	Cycle for treatment or time for duration	Route	Dosage	Gestational age	Birth weight
(g)	Cycle for treatment or time for duration	Outcome	
Tulin Gokmen	2011	Turkey	RCT	102 (52/50)	between January 2009 and February 2010.	oral ibuprofen	10-5-5 mg/kg (initial-24 h–48 h)	28.5 ± 1.9	1,170 ± 297	3 courses	intravenous ibuprofen	10-5-5 mg/kg (initial-24 h–48 h)	28.7 ± 2.1	1,205 ± 366	3 courses	PDA closure rate
surgical rate
BPD
Severe ROP
NEC,
IVH (severe)
Sepsis
Death
AKI
GIH
plasma creatinine	
Omer Erdeve	2012	Turkey	RCT	70 (36/34)	between January 2010 and February 2011.	oral ibuprofen	10-5-5 mg/kg (initial-24 h–48 h)	26.4 ± 1.1	892 ± 117	3 courses	intravenous ibuprofen	10-5-5 mg/kg(initial-24 h–48 h)	26.3 ± 1.3	872 ± 123	3 courses	PDA closure rate
surgical rate
BPD
Severe ROP
NEC,
IVH(severe)
Sepsis
Death
plasma creatinine	
Ahmed Cherif	2008	Tunisia	RCT	64 (32/32)	January 2007 to December 2007	oral ibuprofen	10-5-5 mg/kg(initial-24 h–48 h)	29.3 ± 1.2	1227.2 ± 188	3 courses	intravenous ibuprofen	10-5-5 mg/kg(initial-24 h–48 h)	28.3 ± 1.1	1,197.72 ± 158	3 courses	PDA closure rate
surgical rate
BPD
NEC
IVH(severe)
Sepsis
SIP
PVL	
Priyantha Ebenezer Edison	2021	Singapore	RCT	11 (6/5)	between June 2017 and February 2019	oral ibuprofen	10-5-5 mg/kg(initial-24 h–48 h)	26.4 ± 1.37	821.96 ± 184.93	3 courses	intravenous ibuprofen	10-5-5 mg/kg(initial-24 h–48 h)	27.8 ± 2.5	1,096.26 ± 313.62	3 courses	PDA closure rate
surgical rate
BPD
Severe ROP
NEC,	
Dina Abushanab	2020	Qatar	observational study	99 (40/59)	from 2014 through 2018.	oral ibuprofen	10-5-5 mg/kg(initial-24 h-49 h)	28.36 ± 2.91	1213.6 ± 430.74	3 courses	intravenous ibuprofen	10-5-5 mg/kg(initial-24 h-49 h)	27.1 ± 2.8	1,025 ± 354.8	3 courses	PDA closure rate
surgical rate
Thrombocytopenia
SIP
Sepsis	
Edmond Pistulli	2013	Albania	RCT	68 (36/32)	from January 2010 to December 2012.	oral ibuprofen	10-5-5 mg/kg(initial-24 h-50 h)	N	N	3 courses	intravenous ibuprofen	10-5-5 mg/kg(initial-24 h-50 h)	N	N	3 courses	PDA closure rate
surgical rate
Sepsis	
Ozgur Olukman	2012	Turkey	observational study	66 (24/42)	between April 2009 and June 2010	oral ibuprofen	10-5-5 mg/kg(initial-24 h-51 h)	30.1 ± 3.4	1,424 ± 643	3 courses	intravenous ibuprofen	10-5-5 mg/kg(initial-24 h-51 h)	29.3 ± 3.3	1261 ± 470	3 courses	PDA closure rate
surgical rate
NEC
Thrombocytopenia
Hyperbilirubinemia
SIP
Oliguria
Pulmonary hemorrhage
PPHN	
Ayala Gover	2022	Israel	observational study	39 (19/20)	between January 2012 until June 2020	oral paracetamol	15 mg/kg/ 6 h	27.8 ± 2.8	959.62 ± 176.90	3 to 7days	intravenous paracetamol	15 mg/kg/6 h	27.4 ± 1.9	853.60 ± 89.70	3 to 7days	PDA closure rate BPD
Severe ROP
NEC,
IVH(severe)	
Selim Sancak	2014	Turkey	observational study	18 (8/10)	between January 2013 and June 2014	oral paracetamol	60 mg/kg/day for 3 consecutive days	27.54 ± 2.1	1,074.35 ± 217.89	2 courses	intravenous paracetamol	60 mg/kg/day for 3 consecutive days	26.21 ± 1.1	839.03 ± 119.61	2 courses	PDA closure rate
surgical rate
BPD
death	
Afif El-Khuffash	2014	Canada	observational study	21 (12/9)	between January 2012 and June 2013	oral paracetamol	15 mg/ kg every 6 h	25 ± 1	820 ± 174	2 to 7 days	intravenous paracetamol	15 mg/ kg every 6 h	26 ± 1	822 ± 224	6 days	PDA closure rate	
Nihat Demir	2014	Turkey	RCT	72 (36/36)	between January 2014 and July 2015	oral ibuprofen	10-5-5 mg/kg(initial-24 h-49 h)	30.2 ± 2.04	1,435 ± 343	3 courses	rectal ibuprofen	10-5-5 mg/kg(initial-24 h-49 h)	29.7 ± 2.3	1,330 ± 457	3 courses	PDA closure rate
surgical rate BPD
Severe ROP
NEC,
IVH(severe)
Death
GIH
Sepsis
Surgical rate
plasma creatinine	

Table 2 Level of evidence and Jadad quality score.

First author (year)	Level of evidence	Modified jaded scale	
		Randomization	Blinding	Description of withdrawals and dropouts	Inclusion/ exclusion criteria	Adverse effects	Statistical analysis	Total	
Gokmen (2011)	Ib	2	2	1	1	1	1	8	
Erdeve (2012)	Ib	2	2	1	1	1	1	8	
Cherif (2008)	Ib	2	2	0	1	1	1	7	
Edison (2021)	Ib	2	2	1	1	1	1	8	
Pistulli (2013)	Ib	2	0	1	1	1	1	6	
Demir (2014)	Ib	2	0	0	1	1	1	5	

Table 3 Newcastle-Ottawa scale.

First author
(year)	Newcastle-Ottawa scale	
	Is the case definition adequate?	Representativeness of the cases	Selection of controls	Definition of controls	Comparability	Ascertainment of exposure	Same method of ascertainment for cases and controls	Non-
Response rate	Total score	
Abushanab (2020)	★	★	★	/	★★	★	★	★	8	
Olukman (2012)	★	★	★	/	★★	★	★	/	7	
Gove (2022)	★	★	★	/	★★	★	★	★	8	
Afif El-Khuffash (2014)	★	★	★	/	★★	★	★	/	7	
Selim (2014)	★	★	/	/	★★	★	★	/	6	
Notes.

A forward slash (/) represents “0” score while stars (* and **) represent “1” score and “2” score, respectively.

Quality assessment results for included studies based on the Jadad scale and the NOS method were displayed in Tables 2 and 3.

Outcomes

A total of six RCTs and five observational studies involving 630 premature neonates in total were included for analysis. Within this cohort, 480 neonates were in the ibuprofen group (oral vs. intravenous), 78 in the paracetamol group (oral vs. intravenous), and 72 in the ibuprofen group (rectal vs. oral). All outcomes are shown in Figs. 2–5, Figs. S1–S36, and Table 4.

Primary outcomes

PDA closure rate after the first course of drug therapy.

Six studies reported the PDA closure rate after the first course of treatment in preterm infants who received ibuprofen either orally or intravenously. In these studies, preterm infants received three doses of ibuprofen (10 mg/kg, 5 mg/kg, or 5 mg/kg) once every 24 h. The PDA closure rate was 84% (n = 156/186) for preterm infants taking ibuprofen orally compared to 66% (n = 128/195) for those treated intravenously. A combined meta-analysis of RCTs and observational studies indicated a significant difference in PDA closure rates (RR = 1.27, 95% CI [1.13–1.44]; P < 0.0001, I2 = 0%; Fig. 2A) between the oral ibuprofen and intravenous ibuprofen groups. Further meta-analysis of RCTs (RR = 1.32, 95% CI [1.15–1.51] P<0.0001, I2 = 0%; Fig. 2B) confirmed a statistically significant difference in PDA closure rates between the oral and intravenous ibuprofen groups, while the analysis of observational studies (RR = 1.13, 95% CI [0.88–1.46]; P = 0.35; Fig. 2C) revealed no significant difference between these two groups in PDA closure rates.

The daily dose of paracetamol was 15 mg/kg every 6 h. Two studies provided the PDA closure rate following the first course of treatment in preterm infants treated with oral or intravenous paracetamol. The study by Gover et al., administered paracetamol for 3 to 7 days, while Sancaka et al. administered it was for 3 days. The PDA closure rate was 63% (n = 17/27) for preterm infants receiving oral paracetamol and 40% (n = 12/30) for those receiving it intravenously. Meta-analysis of observational studies (RR = 1.33, 95% CI [0.45–3.95]; P = 0.60, I2 = 55%; Fig. 2D) indicated no difference between oral and intravenous paracetamol in PDA closure rates.

Only one study evaluated the efficacy of rectal versus oral ibuprofen in light of PDA closure rates after the first course of treatment. The rate was 86% (n = 31/36) for preterm infants receiving rectal ibuprofen and 83% (n = 30/36) for those in the oral group. Meta-analysis of RCTs (RR = 1.03, 95% CI [0.85–1.26] P = 0.74; Fig. 3A) showed no significant difference between the two groups in PDA closure rates.

PDA closure rate after the total course of drug therapy.

After failing to receive the first course of ibuprofen treatment, patients with hsPDA were administered a second or third course of drug treatment.

Six studies reported the PDA closure rate after the total course of treatment among preterm infants treated with oral or intravenous ibuprofen. The PDA closure rate was 90% (n = 200/220) for preterm infants receiving oral ibuprofen and 79% (n = 197/249) for those treated intravenously. Combined meta-analysis of RCTs and observational studies indicated no significant difference in PDA closure rates (RR = 1.08, 95% CI [0.96–1.21] P = 0.20, I2 = 85%; Fig. 3B) between oral ibuprofen and intravenous ibuprofen groups. Meta-analysis of RCTs (RR = 1.03, 95% CI [0.96–1.11]; P = 0.37; I2 = 50%; Fig. 3C) and observational studies (RR = 1.35, 95% CI [0.36–5.14]; P = 0.66, I2 = 98%; Fig. 3D) also revealed no significant difference between these two groups regarding PDA closure rates.

Figure 2 (A) Risk difference of combined of RCTS and observational studies of PDA closure rate after the first course of ibuprofen. (B) Risk difference of RCTS of PDA closure rate after the first course of ibuprofen. (C) Risk difference of observational studies of PDA closure rate after the first course of ibuprofen. (D) Risk difference of observational studies of PDA cosure rate after the first course of paracetamol.

Figure 3 (A) Risk difference of RCTS of PDA closure rate after the first course of ibuprofen. (B) Risk difference of combined of RCTS and observational studies of PDA closure rate after total course of ibuprofen. (C) Risk difference of RCTS of PDA closure rate after total course of ibuprofen. (D) Risk difference of observational studies of PDA closure rate after total course of ibuprofen.

Figure 4 (A) Risk difference of observational studies of PDA closure rate after total course of paracetamol. (B) Risk difference of RCTS of PDA closure rate after total course of ibuprofen. (C) Risk difference of RCTs and observational studies of the need for surgical ligation of PDA after the ibuprofen treatment. (D) Risk difference of RCTs of the need for surgical ligation of PDA after the ibuprofen treatment.

Figure 5 (A) Risk difference of observational studies of the need for surgical ligation of PDA after the ibuprofen treatment. (B) Risk difference of observational studies of the need for surgical ligation of PDA after the paracetamol treatment. (C) Risk difference of RCTs of the need for surgical ligation of PDA after the ibuprofen treatment. (D) Risk difference of RCTs of mortality during the ibuprofen therapy.

Two studies provided data on the PDA closure rate after the total course of treatment among preterm infants treated with either oral or intravenous paracetamol. The PDA closure rate was 65% (n = 13/20) for preterm infants receiving oral paracetamol and 79% (n = 15/19) for those in the intravenous group. Meta-analysis of observational studies (RR = 0.86, 95% CI [0.38–1.91]; P = 0.71, I2 = 76%; Fig. 4A) also revealed no significant difference between these two groups in PDA closure rates.

Only one study compared the efficacy of the total course of treatment with rectal or oral ibuprofen in light of PDA closure rates. The PDA closure rate was 94% (n = 34/36) for preterm infants receiving rectal ibuprofen and 94% (n = 34/36) for those in the oral group. Meta-analysis of the RCT (RR = 1.00, 95% CI [0.89–1.12] P = 1.00; Fig. 4B) indicated no significant difference between rectal and intravenous ibuprofen group in PDA closure rates.

Need for surgical ligation of PDA after the allocated treatment.

Six studies contributed to the analysis of surgical ligation rates among preterm infants after they were treated with oral or intravenous ibuprofen. The surgical ligation rate was 1% (n = 3/228) for preterm infants receiving oral ibuprofen and 4% (n = 12/249) for those in the intravenous group. Combined meta-analysis of RCTs and observational studies indicated no significant difference surgical ligation rate (RR = 0.39, 95% CI [0.13–1.18]; P = 0.09, I2 = 0%; Fig. 4C) between oral ibuprofen and intravenous ibuprofen group. A separate meta-analysis of RCTs (RR = 0.40, 95% CI [0.11–1.44]; P = 0.16, I2 = 0%; Fig. 4D) and observational studies (RR = 0.37, 95% CI [0.04–3.25] P = 0.37, I2 = 0%; Fig. 5A) also revealed no significant difference between these two groups in surgical ligation rate.

One study provided the surgical ligation rate among preterm infants after they were treated with oral or intravenous paracetamol. The surgical ligation rate was 13% (n = 1/8) for preterm infants receiving oral paracetamol and 10% (n = 1/10) for those in the intravenous group. Meta-analysis of observational studies (RR = 1.25, 95% CI [0.09–17.02], P = 0.87; Fig. 5B) also showed no significant difference between oral and intravenous paracetamol groups in the surgical ligation rate.

Only one study compared the surgical ligation rate among preterm infants treated with rectal or oral ibuprofen. The surgical ligation rate was 5% (n = 2/36) for preterm infants taking rectal ibuprofen and 5% (n = 2/36) for those in the oral group. Meta-analysis of RCTs (RR = 1.00, 95% CI [0.15–6.72] P = 1.00; Fig. 5C) showed that there was no significant difference between patients taking rectal or intravenous ibuprofen in surgical ligation rate.

Table 4 The outcome of meta-analysis.

Outcome	Type of study	Oral
(events/total)	Intravenous (events/total)	RR, 95% CI, P, I2	
Ibuprofen	
PDA closure rate after the first course of Ibuprofen	Combined of RCTs and observational studies	156/186	128/195	RR = 1.27, 95% CI [1.13–1.44], P < 0.0001, I2= 0%	
	RCTs	136/162	97/153	RR = 1.32, 95% CI [1.15–1.51], P < 0.0001, I2= 0%	
	observational studies	20/24	31/34	RR = 1.13, 95% CI [0.88–1.46], P = 0.35	
PDA closure rate after the total course of Ibuprofen					
	Combined of RCTs and observational studies	200/220	197/249	RR = 1.08, 95% CI [0.96–1.21], P = 0.20, I2= 85%;	
	RCTs	150/156	135/148	RR = 1.03, 95% CI [0.96–1.11], P = 0.37, I2= 50%	
Need for surgical ligation of PDA after the allocated treatment	observational studies	50/64	62/101	RR = 1.35, 95% CI [0.36–5.14], P = 0.66, I2= 98%	
	combined meta-analysis of RCTs and observational studies	3/228	12/249	RR = 0.39, 95% CI [0.13–1.18], P = 0.09, I2= 0%	
	RCTs	3/156	9/148	RR = 0.40, 95% CI [0.11–1.44], P = 0.16, I2= 0%	
Mortality during the therapy of ibuprofen	observational studies	0/72	3/101	RR = 0.37, 95% CI [0.04–3.25], P = 0.37, I2= 0%	
Incidence of BPD/CLD					
	RCTs	4/88	2/84	RR = 1.91, 95% CI [0.36–10.12], I2= 0%P=0.45	
Severe ROP					
Incidence of NEC	RCTs	34/126	37/121	RR = 0.88, 95% CI [0.61–1.26], P = 0.48	
	RCTs	11/94	15/89	RR = 0.68, 95% CI [0.34–1.38], I2= 0%, P = 0.28	
Incidence of IVH	combined meta-analysis of RCTs and observational studies	11/184	13/217	RR = 0.97, 95% CI [0.44–2.12], I2= 0%, P = 0.93	
	RCTs	8/120	9/116	RR = 0.87, 95% CI [0.35–2.21], I2= 0%, P = 0.78	
Incidence of localized bowel perforation/SIP	observational studies	3/64	4/101	RR = 1.25, 95% CI [0.29–5.34], I2= 0%, P = 0.77	
	RCTs	28/126	24/121	RR = 1.11, 95% CI [0.70–1.76], I2= 0%, P = 0.65	
Incidence of GIH	combined meta-analysis of RCTs and observational studies	2/70	4/106	RR = 0.87, 95% CI [0.18–4.22], I2= 0%, P = 0.86	
	RCTs	1/6	0/5	RR = 2.57, 95% CI [0.13–52.12], I2= 0%, P = 0.54	
Incidence of sepsis	observational studies	1/64	4/101	RR = 0.58, 95% CI [0.09–3.68], P = 0.56	
Incidence of PVL	RCTs	1/52	0/50	RR = 2.89, 95% CI [0.12–69.24], P = 0.51	
Incidence of oliguria	RCTs	35/156	43/148	RR = 0.77, 95% CI [0.52–1.12], I2= 0%, P = 0.17	
Incidence of pulmonary hemorrhage	RCTs	2/32	2/32	RR = 1.00, 95% CI [0.15–6.67], P = 1.00	
Incidence of persistent pulmonary hypertension	observational studies	2/24	5/42	RR = 0.70, 95% CI [0.15–3.34], P = 0.65	
Incidence of thrombocytopenia	observational studies	2/24	3/42	RR = 1.17, 95% CI [0.21–6.50], P = 0.86	
Incidence of hyperbilirubinemia	observational studies	2/24	1/42	RR = 3.50, 95% CI [0.33–36.61], P = 0.30	
	observational studies	4/64	7/101	RR = 1.05, 95% CI [0.35–3.13], P = 0.93	
	observational studies	1/24	8/42	RR = 0.22, 95% CI [0.03–1.64], P = 0.14	
Paracetamol	
PDA closure rate after the first course of paracetamol	observational studies	17/27	12/30	RR 1.33, 95% CI [0.45–3.95], P = 0.60, I2= 55%	
	observational studies	13/20	15/19	RR = 0.86, 95% CI [0.38–1.91], P = 0.71, I2= 76%	
PDA closure rate after the total course of paracetamol	observational studies	1/8	1/10	RR = 1.25, 95% CI [0.09–17.02], P = 0.87	
Need for surgical ligation of PDA after the allocated treatment	observational studies	1/8	3/8	RR = 0.42, 95% CI [0.05–3.28], P = 0.41	
Mortality during the therapy of paracetamol	observational studies	22/27	25/30	RR = 1.03, 95% CI [0.12–8.75], P = 0.97	
Incidence of BPD/CLD	observational studies	9/27	8/30	RR = 1.26, 95% CI [0.57–2.79], I2= 0%, P = 0.57	
Severe ROP	observational studies	4/19	3/20	RR = 1.40, 95% CI [0.36–5.4], P = 0.62	
Incidence of NEC	observational studies	8/19	11/20	RR = 0.77, 95% CI [0.40–1.48], P = 0.43	
Incidence of IVH					
Ibuprofen	
PDA closure rate after the first course of Ibuprofen	RCTs	136/162	97/153	RR = 1.32, 95% CI [1.15–1.51], P < 0.0001, I2= 0%;	
PDA closure rate after the total course of Ibuprofen	RCTs	150/156	135/148	RR = 1.03, 95% CI [0.96–1.11], P = 0.37, I2= 50%	
Need for surgical ligation of PDA after the allocated treatment	RCTs	3/156	9/148	RR = 0.40, 95% CI [0.11–1.44], P = 0.16, I2= 0%	
Mortality during the therapy of ibuprofen	RCTs	4/88	2/84	RR = 1.91, 95% CI [0.36–10.12], I2= 0% P = 0.45	
Incidence of BPD/CLD	RCTs	34/126	37/121	RR = 0.88, 95% CI [0.61–1.26], P = 0.48	
Severe ROP	RCTs	11/94	15/89	RR = 0.68, 95% CI [0.34–1.38], I2= 0%, P = 0.28	
Incidence of NEC	RCTs	8/120	9/116	RR = 0.87, 95% CI [0.35–2.21], I2= 0%, P = 0.78	
Incidence of IVH	RCTs	28/126	24/12	RR = 1.11, 95% CI [0.70–1.76], I2= 0%, P = 0.65	
Incidence of localized bowel perforation/SIP	RCTs	1/6	0/5	RR = 2.57, 95% CI [0.13–52.12], I2= 0%, P = 0.54	
Incidence of GIH RCTs	1/52	0/50		RR = 2.89, 95% CI [0.12–69.24], P = 0.51	
Incidence of sepsis	RCTs	35/156	43/148	RR = 0.77, 95% CI [0.52–1.12], I2= 0%, P = 0.17	
Incidence of PVL	RCTs	2/32	2/32	RR = 1.00, 95% CI [0.15–6.67], P = 1.00	

Secondary outcomes

Mortality during the therapy of COX/POX inhibitors.

Four studies explored mortality among preterm infants who took ibuprofen (oral vs. intravenous), paracetamol (oral vs. intravenous), or ibuprofen (rectal vs. oral).

Meta-analysis of RCTs (RR = 1.91, 95% CI [0.36–10.12], I2 = 0%; P = 0.45; Fig. 5D) showed that there was no significant difference in mortality between the oral and intravenous ibuprofen group.

Meta-analysis of observational studies (RR = 0.42, 95% CI [0.05–3.28]; P = 0.41; Fig. S1) showed that there was no significant difference in mortality between the oral and intravenous paracetamol groups.

Meta-analysis of RCT (RR = 1.00, 95% CI [0.15–6.72] P = 1.00; Fig. S2) revealed no significant difference in mortality between the rectal and oral ibuprofen group.

Incidence of BPD/CLD.

Seven studies contributed to the analysis of the incidence of BPD or CLD among preterm infants taking ibuprofen (oral vs. intravenous), paracetamol (oral vs. intravenous), or ibuprofen (rectal vs. oral).

Meta-analysis of RCTs (RR = 0.88, 95% CI [0.61–1.26] P = 0.48) revealed no significant difference between the oral and intravenous ibuprofen group with regard to the incidence of BPD or CLD (Fig. S3).

Meta-analysis observational studies (RR = 1.03, 95% CI [0.12–8.75] P = 0.97; Fig. S4) showed that no statistically significant difference emerged between oral and intravenous paracetamol group with regard to the incidence of BPD or CLD.

Meta-analysis of RCTs (RR = 0.83, 95% CI [0.28–2.49] P = 0.74; Fig. S5) showed that there was no significant difference between rectal and oral ibuprofen group with regard to the incidence of BPD/CLD.

Severe ROP.

Six studies provided data on the incidence of severe ROP among preterm infants treated with ibuprofen (oral vs intravenous), paracetamol (oral vs intravenous), or ibuprofen (rectal vs oral).

Meta-analysis of RCTs (RR = 0.68, 95% CI [0.34–1.38] I2 = 0%, P = 0.28) indicated no significant difference between the oral and intravenous ibuprofen group with regard to the incidence of severe ROP (Fig. S6).

Meta-analysis of observational studies (RR = 1.26, 95% CI [0.57–2.79] I2 = 0%, P = 0.57; Fig. S7) showed that there was no significant difference between oral and intravenous paracetamol group regarding the incidence of severe ROP.

Meta-analysis of RCT (RR = 0.50, 95% CI [0.05–5.27] P = 0.56; Fig. S8) showed that there was no significant difference between the rectal and oral ibuprofen group concerning the incidence of severe ROP.

Incidence of NEC.

Seven studies investigated the incidence of NEC among preterm infants taking ibuprofen (oral vs intravenous), paracetamol (oral vs intravenous), or ibuprofen (rectal vs oral).

Meta-analysis of RCTs (RR = 0.87, 95% CI [0.35–2.21] I2 = 0%, P = 0.78; Fig. S9) and observational studies (RR = 1.25, 95% CI [0.29–5.34] I2 = 0%, P = 0.77; Fig. S10) revealed no statistically significant difference between oral and intravenous ibuprofen group regarding the incidence of NEC. The combined meta-analysis of cohort studies and RCTs demonstrated no significant difference between the oral and intravenous ibuprofen group in light of the incidence of NEC (RR = 0.97, 95% CI [0.44–2.12] I2 = 0%, P = 0.93; Fig. S11).

Similarly, for the oral vs. intravenous paracetamol comparison, the meta-analysis of observational studies (RR = 1.40, 95% CI [0.36–5.46] P = 0.62; Fig. S12) found no significant difference in the incidence of NEC.

Regarding rectal vs. oral ibuprofen, the analysis of RCTs (RR = 0.67, 95% CI [0.12–3.75] P = 0.65; Fig. S13) indicated no significant difference in the incidence of NEC.

Incidence of IVH.

Six studies reported the incidence of IVH during the administration of ibuprofen (oral vs. intravenous), paracetamol (oral vs. intravenous), or ibuprofen (rectal vs. oral).

Meta-analysis of RCTs (RR = 1.11, 95% CI [0.70–1.76] I2 = 0%, P = 0.65; Fig. S14) showed that there was no significant difference between oral and intravenous ibuprofen group in view of the incidence of IVH.

Meta-analysis of observational studies (RR = 0.77, 95% CI [0.40–1.48] P = 0.43; Fig. S15) demonstrated that no significant difference emerged between the oral and intravenous paracetamol group with regard to the incidence of IVH.

For the comparison of rectal vs. oral ibuprofen, the meta-analysis of RCTs (RR = 0.57, 95% CI [0.18–1.78] P = 0.34; Fig. S16) showed no significant difference in the incidence of IVH.

Incidence of localized bowel perforation/SIP.

Three studies provided data on the incidence of localized bowel perforation/SIP during the administration of ibuprofen (oral vs intravenous).

Meta-analysis of RCTs (RR = 2.57, 95% CI [0.13–52.12] I2 = 0%, P = 0.54; Fig. S17) and observational studies (RR = 0.58, 95% CI [0.09–3.68] P = 0.56; Fig. S18) indicated no significant difference between oral and intravenous ibuprofen group regarding the incidence of localized bowel perforation/SIP. The combined meta-analysis of cohort studies and RCTs demonstrated no significant difference between the oral and intravenous ibuprofen group with regard to the incidence of localized bowel perforation/SIP (RR = 0.87, 95% CI [0.18–4.22] I2 = 0%, P = 0.86; Fig. S19).

Incidence of GIH.

Two studies investigated the incidence of GIH among preterm infants taking ibuprofen (oral vs intravenous) or ibuprofen (rectal vs oral).

Meta-analysis of RCTs (RR = 2.89, 95% CI [0.12–69.24] P = 0.51; Fig. S20) showed that there was no significant difference in the incidence of GIH between the oral and intravenous ibuprofen groups.

Meta-analysis of RCTs (RR = 0.33, 95% CI [0.01–7.92] P = 0.50; Fig. S21) revealed no significant difference in the incidence of GIH between the rectal and oral ibuprofen groups.

Incidence of sepsis.

Five studies provided data on the incidence of sepsis during the administration of ibuprofen (oral vs intravenous) or ibuprofen (rectal vs. oral).

Meta-analysis of RCTs (RR = 0.77, 95% CI [0.52–1.12] I2 = 0%, P = 0.17; Fig. S22) revealed no significant difference in the incidence of sepsis between the oral and intravenous ibuprofen groups.

Meta-analysis of RCTs (RR = 0.83, 95% CI [0.28–2.49] P = 0.74; Fig. S23) demonstrated no significant difference in the incidence of sepsis between the rectal and oral ibuprofen groups.

Incidence of PVL.

One study provided data on the incidence of PVL during the administration of ibuprofen (oral vs intravenous).

Meta-analysis of the RCTs (RR = 1.00, 95% CI [0.15–6.67] P = 1.00; Fig. S24) indicated no significant difference in the incidence of PVL between the oral and intravenous ibuprofen groups.

Incidence of oliguria.

One study investigated the incidence of oliguria among preterm infants taking ibuprofen (oral vs intravenous).

Meta-analysis of the observational study (RR = 0.70, 95% CI [0.15–3.34] P = 0.65; Fig. S25) showed that there was no significant difference between the oral and intravenous ibuprofen groups with regard to the incidence of oliguria.

Incidence of pulmonary hemorrhage.

One study reported the incidence of pulmonary hemorrhage during the administration of ibuprofen (oral vs intravenous).

Meta-analysis of the observational study (RR = 1.17, 95% CI [0.21–6.50] P = 0.86; Fig. S26) demonstrated no significant difference between the oral and intravenous ibuprofen groups regarding the incidence of pulmonary hemorrhage.

Incidence of persistent pulmonary hypertension.

One study reported data on the incidence of persistent pulmonary hypertension during the administration of ibuprofen (oral vs. intravenous).

Meta-analysis of the observational study (RR = 3.50, 95% CI [0.33–36.61] P = 0.30; Fig. S27) indicated no significant difference in the incidence of persistent pulmonary hypertension between the oral and intravenous ibuprofen groups.

Incidence of thrombocytopenia.

Two studies investigated the incidence of thrombocytopenia among preterm infants taking ibuprofen (oral vs intravenous).

Meta-analysis of observational studies (RR = 1.05, 95% CI [0.35–3.13] P = 0.93; Fig. S28) demonstrated that no significant difference between the oral and intravenous ibuprofen groups with regard to the incidence of thrombocytopenia.

Plasma creatinine.

Three studies examined changes in plasma creatinine levels among preterm infants taking ibuprofen (oral vs. intravenous) or ibuprofen (rectal vs. oral).

Meta-analysis of related studies (WMD = 0.05, 95% CI [−0.06–0.16] I2 = 35%, P = 0.35; Fig. S29) demonstrated no significant difference in plasma creatinine levels between preterm infants before and after taking ibuprofen orally (WMD = − 0.01, 95% CI [−0.10 to −0.08], P = 0.87; Fig. S30) and no statistical difference emerged in the level of plasma creatinine between preterm infants before and after taking ibuprofen intravenously.

Meta-analysis of related studies (WMD = − 0.01, 95% CI [−0.15–0.13] I2 = 0%, P = 0.89; Fig. S31) showed that there was no significant difference in the level of plasma creatinine between preterm infants before and after taking ibuprofen orally (WMD = −0.01, 95% CI [−0.10–0.08]]; P = 0.87; Fig. S32) and no statistical difference between emerged in the level of plasma creatinine between preterm infants before and after taking ibuprofen rectally.

Plasma bilirubin.

Three studies provided data on changes in the level of plasma bilirubin during the administration of ibuprofen (oral vs intravenous), paracetamol (oral vs intravenous), or ibuprofen (rectal vs oral), respectively.

Meta-analysis of the study investigating oral vs. intravenous ibuprofen (WMD = − 0.03, 95% CI [−0.87–0.81] I2 = 0%, P = 0.95; Fig. S33) revealed no significant difference in the level of plasma bilirubin between preterm infants before and after taking ibuprofen orally (WMD = 0.37, 95% CI [−0.42–1.15] P = 0.36; Fig. S34) and no statistical difference in the level of plasma bilirubin between preterm infants before and after taking ibuprofen intravenously.

For the comparison of rectal vs. oral ibuprofen (WMD = 0.50, 95% CI [−0.78–1.78] P = 0.44; Fig. S35) demonstrated that there was no statistically significant difference in the level of plasma bilirubin between preterm infants before and after taking ibuprofen orally (WMD = 0.31, 95% CI [−0.94–1.56] P = 0.63; Fig. S36) and no statistical difference emerged in the level of plasma bilirubin between preterm infants before and after taking ibuprofen rectally.

Incidence of hyperbilirubinemia.

One study contributed to the analysis of the incidence of hyperbilirubinemia during the administration of ibuprofen (oral vs. intravenous), paracetamol (oral vs. intravenous), or ibuprofen (rectal vs. oral).

Meta-analysis of the observational study (RR = 0.22, 95% CI [0.03–1.64] P = 0.14; Fig. S37) showed that there was no statistical difference between oral and intravenous ibuprofen group with regard to the incidence of hyperbilirubinemia.

Risk of bias assessment.

Figures A1–A27, B1–B27, present the overall and individual results of the risk of bias assessments. The sensitivity analysis confirmed the stability of our analysis results and no publication bias was identified. Therefore, there is no need for further elimination or revision of these studies.

Discussion

This meta-analysis investigated how routes of administration impacted the safety and effectiveness of medications for treating premature infants with PDA. In many countries, ibuprofen or paracetamol is frequently administered orally owing to its convenience and cost-effectiveness. We included six RCTs and two observational studies investigating the use of ibuprofen, along with three observational studies focusing on paracetamol administration. With more cases included in our study compared to previous meta-analyses, our analysis results showed that oral administration of ibuprofen or paracetamol outperformed intravenous administration to a certain extent regarding the effectiveness in closing PDA, without increasing the risk of adverse events. Over a decade ago, intravenous medications were the standard for treating premature infants due to concerns about the high osmolarity of oral medication, which may increase the risk of feeding intolerance and gastrointestinal disorder (Gouyon & Kibleur, 2010). However, recent studies have shown that oral ibuprofen or acetaminophen can effectively close PDA for premature infants with fewer adverse effects compared with intravenous administration (Cherif et al., 2008; Gover et al., 2022). A recent Cochrane systematic review, which focused on five RCTs comparing the efficacy and adverse effects associated with oral or intravenous ibuprofen in closing hsPDA (Ohlsson, Walia & Shah, 2003), indicated that the risk of PDA closure failure with oral ibuprofen was lower than that with intravenous ibuprofen, whether administered as single dose or three doses (typical RR = 0.38, 95% CI [0.26–0.56] moderate-quality evidence). Moreover, there were no significant differences in adverse effects between these two administration methods. Our study included additional research (two observational studies published in 2012 and 2020, and two RCTs published in 2014 and 2021). Different from the Cochrane systematic review, we excluded the study by Akar et al. (2017) due to its similarity to the study by Gokmen et al. (2011) in terms of study timeframes, patient characteristics, drug interventions for PDA treatment, and PDA closure rates. Our findings aligned with the abovementioned Cochrane systematic review, emphasizing the superior efficacy of oral ibuprofen compared to intravenous one in closing PDA after the first course of treatment. In contrast to the study by Olukman et al. (2012) and Pistulli et al. (2014), we found that the premature infants included in their study exhibited higher birth weights and larger gestational ages compared to the subjects in the other four studies. This variance in demographics may explain that infants with lower birth weights or smaller gestational ages had a preference for the oral route of administration. However, this assumption lacks concrete evidence and warrants verification in future research. Surgical ligation serves as a last-resort option when drug interventions fail to close the PDA, indirectly indicating the efficacy of both oral and intravenous administration methods. Interestingly, our findings diverge from those of the Cochrane systematic review mentioned earlier, which found no difference in PDA closure rates between oral and intravenous administration when considering the entire treatment course, leading to the necessity of surgical ligation. The mechanism underpinning the enhanced efficacy of oral ibuprofen remains unclear. Some researchers have suggested that slower absorption and a longer plasma half-life of oral ibuprofen may lead to more prolonged drug exposure to ductal tissue (Erdeve et al., 2012; Sancak et al., 2016; Jensen, Kampmann & Andersen, 2020). Further revealed that the first pass effect during oral ibuprofen administration transforms inactive R ibuprofen into S ibuprofen. Given that ibuprofen was a racemic mixture of inactive R and active S mirror-image enantiomers, this suggests that oral ibuprofen administration contributes to a higher concentration of active ibuprofen compared to intravenous administration. However, the study conducted by Smit et al. (2023) contradicts this notion showing that the concentration of S-ibuprofen associated with oral ibuprofen was lower than that with intravenous administration for a significant portion of the dosing interval in preterm infants with hsPDA. This indicates that delayed absorption following oral ibuprofen administration may not lead to increased S-ibuprofen concentration (Smit et al., 2023). Also noted that the higher peak concentration of ibuprofen after intravenous administration might lead to decreased glomerular filtration rates, increased fluid load, and reduced PDA closure rates. Another recent Cochrane systematic review (Ohlsson, Walia & Shah, 2020) confirmed that the plasma cystatin C and creatinine levels after ibuprofen was orally administered were lower than those after intravenous administration of ibuprofen.

A Cochrane systematic analysis revealed no significant difference between paracetamol and ibuprofen or indomethacin for closing PDA among premature infants. Given that paracetamol is often associated with fewer adverse effects compared with ibuprofen or indomethacin, it has emerged as a preferred choice for alternative or rescue therapy when ibuprofen or indomethacin was not suitable for preterm infants with hsPDA (Hamrick et al., 2020). There is currently a dearth of RCTs directly comparing the effects of orally or intravenously administered paracetamol. A recent network meta-analysis showed that the odds ratio(OR) of oral paracetamol in closing PDA (OR = 14.18; 95% CI [4.73–42.53]) was higher than that of intravenous paracetamol (OR = 6.70; 95% CI [1.31–34.33]) among preterm infants compared with the placebo group, indicating that oral paracetamol outperformed intravenous paracetamol to close PDA (Olowoyeye et al., 2022). Nonetheless, only two retrospective studies have hitherto compared the efficacy of oral and intravenous paracetamol to close hsPDA in preterm infants (Gover et al., 2022; Sancak et al., 2016). Our meta-analysis, in line with current evidence, revealed that PDA closure rates were comparable between patients who received orally administered paracetamol and those taking it intravenously. Importantly, we found no significant differences in the risk of adverse events between the oral and intravenous paracetamol groups. Animal studies have shown that the effect of paracetamol on ductus constriction is closely correlated with drug concentration (El-Khuffash et al., 2014). In a retrospective study, Bin-Nun et al. (2018) found drug concentration ≥20 mg/L at 4 h after paracetamol was orally administered to preterm infants with hsPDA could predict PDA closure. Conversely, Liebowitz et al. (2019) found that the trough concentration of paracetamol was remarkably similar between oral and intravenous administration, and the contraction of PDA was not correlated to drug concentration. The need for RCTs to directly compare the efficacy of oral and intravenous paracetamol in the treatment of hsPDA among preterm infants is evident and crucial for further elucidating the most effective treatment approach.

Our study showed that there was no difference in the risk of adverse events between preterm infants with PDA whether they received ibuprofen or paracetamol through oral or intravenous routes. Several previous studies have reported gastrointestinal adverse events associated with ibuprofen or acetaminophen. These gastrointestinal issues are chiefly attributed to the high osmolarity of the drug. A striking case report outlines an incident wherein a premature infant developed spontaneous intestinal perforation after the oral administration of paracetamol with exceedinghly high osmolality (Yurttutan & Güllü, 2021). In the realm of oral medication for preterm infants, an osmolality range of 330–350 mOsm/kg H2O is considered appropriate (Tuteja et al., 2020). In our study, most of the included studies concerning ibuprofen reported the use of oral or rectal preparations with osmolalities between 300 and 400 mOsm/kg H2O. However, the included studies investigating acetaminophen did not provide details regarding the osmolality of their oral formulations. In many countries, oral preparations of oral ibuprofen and paracetamol preparation have osmolality levels as high as 4,000–7,000 mOsm/kg H2O, rendering them unsuitable for direct administration in preterm infants (Fernández Polo et al., 2007). Numerous previous clinical studies on closing PDA with oral ibuprofen did not report the dilution method or osmolality of oral ibuprofen preparations. This omission could potentially account for a higher incidence of gastrointestinal bleeding and NEC when compared to placebo or intravenous medication group (Gouyon & Kibleur, 2010; Ohlsson, Walia & Shah, 2010). Shen et al. (2021) uncovered that oral ibuprofen for the treatment of PDA might prolong the duration of partial parenteral nutrition, increase hospital stay, and elevate the risk of cholestasis in infants with low birth weight. Unfortunately, their study, like others, omitted information the osmolality and dilution method employed for the oral ibuprofen preparation. Hence, it is imperative for future research to employ oral ibuprofen or paracetamol preparations with appropriate osmolality suitable for premature infants to mitigate the risk of adverse drug effects. There is currently no research indicating a direct impact of the oral or intravenous route on the occurrence of other adverse events (like BPD, ROP, IVH, AKI, and thrombocytopenia). However, studies have shown that the use of COX inhibitors can indeed exert an impact on the incidence of certain adverse events like BPD and thrombocytopenia. For instance, in the investigation conducted by Raju et al. (2000), ibuprofen demonstrated a trend toward reducing the occurrence and severity of BPD, potentially attributable to its ant-inflammation effects and its capacity to reduce daily mean airway pressures. Conversely, another study suggested that COX inhibitors might not consistently inhibit prostaglandin synthesis, potentially causing thrombocytopenia in premature infants (Brunner et al., 2013). In our study, the analysis results revealed potential advantages associated with the oral administration of ibuprofen or paracetamol over their intravenous counterparts in closing PDA. Nevertheless, it is imperative to acknowledge that further exploration is imperative to forge a comprehensive understanding of the intricate interplay between the oral and intravenous routes and the associated adverse events linked to hs-PDA. Drawing from our findings, it is plausible to posit a latent preference for the oral route to mitigate the risk of accompanying ADE.

With regard to the long-term consequences of oral and intravenous ibuprofen on the prognosis of preterm infants, scant studies have ventured into this domain. Among these studies, only one, characterized by a limited sample size, compared the neurodevelopmental outcomes in infants at 18 to 24 months of corrected age who received oral or intravenous ibuprofen, and no differences were found between the two groups (Eras et al., 2013).

Our study introduces two notable innovations. Firstly, in previous systematic meta-analyses, the assessment of the hs-PDA closure rate for ibuprofen merely encompassed the first course of administration, yielding results in favor of oral ibuprofen wins. However, upon data extraction and analysis, disparities surfaced between the reported hs-PDA closure rate after the first course and the total course in relevant studies. Consequently, in our evaluation of ibuprofen, we compared the PDA closure rate after both the first course and the entire drug therapy course for each route of administration. Our findings revealed a significant difference in hs-PDA closure after the first treatment course. Intriguingly, after accounting for the entire course of drug therapy, no statistically significant difference emerged between oral and intravenous administration, thereby deviating from prior research outcomes. Secondly, our study places an emphasis on the utilization of paracetamol in PDA closure therapy. Several studies have illuminated that paracetamol can effectively and safely close PDA of the premature in premature infants while incurring fewer adverse events compared to ibuprofen or indomethacin (Balachander et al., 2020; Katsaras et al., 2022; Karabulut & Paytoncu, 2019; Nishizaki et al., 2020; Guimarães et al., 2009).

This implies that paracetamol has the potential to serve as a supplementary medication when ibuprofen proves ineffective in closing PDA or when infants struggle to tolerate ibuprofen. Nonetheless, it has not garnered widespread usage as the primary medication for PDA treatment among preterm infants (Jasani, Mitra & Shah, 2022). This is primarily due to the majority of neonates included in paracetamol studies being moderate preterm infants, with limited research pertaining to very preterm infants. Furthermore, no prior meta-analysis has been conducted to compare the safety and efficacy of oral versus intravenous paracetamol as the first-line drug for PDA closure. Given the paucity of studies and limited sample sized currently available, we refrain from making definitive recommendations regarding the choice between intravenous and oral administration of paracetamol for PDA treatment in premature infants.

Our study has two limitations. Firstly, it grapples with the absence of suitable studies investigating the adverse events associated with the oral and intravenous administration of indomethacin and paracetamol. Secondly, our study faces constraints due to the limited number of included studies, stemming from the scarcity of published data within this specific domain. Therefore, there is a need for more research focusing on this respect.

Conclusion

In conclusion, despite certain results in our study not reaching statistical difference, our meta-analysis demonstrated that oral ibuprofen and acetaminophen were associated with minimal adverse effects when used for PDA closure, and it highlighted the ability of premature infants to tolerate oral preparations effectively. These insights have the potential to inform clinical practice, suggesting that the convenience of drug administration and the cost-effectiveness of oral administration may render it a superior option for preterm infants with hsPDA. However, it is essential to exercise caution regarding the osmolality of the administered drug and to dilute it as necessary when administering oral therapy for PDA in premature infants.

Supplemental Information

Supplemental Information 1 Supplementary Materials

Click here for additional data file.

Supplemental Information 2 PRISMA checklist

Click here for additional data file.

Supplemental Information 3 Systematic review registration

Click here for additional data file.

Supplemental Information 4 The rationale of this review

Click here for additional data file.

Additional Information and Declarations

Competing Interests

Author Contributions

Data Availability

The authors declare there are no competing interests.

Hanwen Luo conceived and designed the experiments, performed the experiments, analyzed the data, prepared figures and/or tables, authored or reviewed drafts of the article, and approved the final draft.

Jianghua He performed the experiments, analyzed the data, prepared figures and/or tables, and approved the final draft.

Xiaoming Xu performed the experiments, analyzed the data, prepared figures and/or tables, and approved the final draft.

Hongju Chen conceived and designed the experiments, authored or reviewed drafts of the article, and approved the final draft.

Jing Shi conceived and designed the experiments, authored or reviewed drafts of the article, and approved the final draft.

The following information was supplied regarding data availability:

The search methods are available in the Supplementary File.

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
