# Peer review of "The impact of the route of administration on the efficacy and safety of the drug therapy for patent ductus arteriosus in premature infants: a systematic review and meta-analysis"

_PeerJ, doi:10.7717/peerj.16591_

## Round 0.1 · original submission · Major Revisions

The reviewers raised some critical questions that help improve the quality of the manuscript.

**Language Note:** The review process has identified that the English language must be improved. PeerJ can provide language editing services - please contact us at [email protected] for pricing (be sure to provide your manuscript number and title). Alternatively, you should make your own arrangements to improve the language quality and provide details in your response letter. – PeerJ Staff

Reviewer 1 ·

Basic reporting

In this comprehensive study, the authors have employed rigorous meta-analysis and thorough literature review methods to investigate the safety and effectiveness of oral ibuprofen or paracetamol in the management of PDA in premature infants. Their findings suggest that these interventions may offer distinct advantages in PDA closure without an associated increase in the risk of adverse events, which is a valuable contribution to the field. However, there remain important questions that need to be addressed.

Experimental design

The submission should clearly define the research question, which must be relevant and meaningful. The knowledge gap being investigated should be identified, and statements should be made as to how the study contributes to filling that gap.

Validity of the findings

One notable area for improvement is in referencing the published literature and accurately expressing the 95% confidence intervals (95% CI) to ensure the precision of the results reported. This would enhance the credibility and clarity of the findings.
Furthermore, the manuscript would benefit from a more balanced presentation of visual materials. While there is a dearth of figures in the main body of the manuscript, there appears to be an overabundance of supplementary images. Striking a better balance between the two would help in conveying the key results more effectively.
Regarding the Materials and Methods section, it is advisable to clearly distinguish between included and excluded studies. Separating these two components would enhance the transparency and organization of the research process.

Additional comments

One significant concern pertains to the standard of English in the manuscript. There are instances where the language is challenging to comprehend, which compromises the overall quality of the manuscript. We strongly recommend that the authors seek assistance from a fluent English speaker or enlist the services of a professional language editing service. Addressing both language issues and scientific merit is crucial to ensure the manuscript's clarity and scholarly value.

Reviewer 2 ·

Basic reporting

This paper undertakes the crucial task of evaluating the efficacy and potential adverse events associated with various routes of administration for the treatment of patent ductus arteriosus in premature infants. While the study is a commendable effort, there are several areas that require thoughtful consideration and refinement:

1. To enhance the overall impact and engagement of the manuscript, it is imperative that the authors elaborate on the novelty and innovative potential of their work when compared to the existing literature. This should be articulated more extensively within the abstract and the discussion section, providing readers with a clearer understanding of the unique contributions made by this study.

2. In the sections pertaining to materials and methods, data extraction, and meta-analysis, it is noted that sensitivity analyses and publication bias analyses were conducted but not incorporated into the results section. It is highly advisable to include the outcomes of these analyses to complete the methodological picture and ensure transparency.

3. The citations and reference formatting do not adhere to the prescribed requirements. The authors should diligently review and align their referencing style with the specified guidelines to maintain scholarly rigor and consistency.

4. Figure 1's results should be further expanded upon to provide a comprehensive interpretation, enhancing the visual representation of the data and its significance.

5. The consistency of the notation for I2 should be upheld throughout the manuscript to avoid any confusion or ambiguity.

6. Despite the lack of statistically significant differences in many of the conclusions drawn, it is essential that the authors elucidate the overarching purpose of their paper and its potential implications for future clinical practice. Clarifying the clinical relevance and utility of the findings will add depth to the study's significance.

7. The manuscript exhibits significant deficiencies in the standard of English, which at times hinder comprehension. To rectify this, it is strongly recommended that the authors seek assistance from individuals proficient in the English language or enlist the services of a professional language editing service. This comprehensive language review should encompass not only language improvement but also enhancement of the manuscript's overall scientific merit and coherence.

Experimental design

no comment

Validity of the findings

no comment

Additional comments

no comment

Reviewer 3 ·

Basic reporting

In the manuscript titled " The impact of the route of administration on the efficacy and adverse events in the drug therapy of the Patent ductus artiriousus in the premature infants: a systematic meta - analysis and literature review" the authors have conducted a comprehensive investigation, shedding light on the potential advantages of utilizing oral ibuprofen or paracetamol as a prospective therapeutic strategy for closing PDA in premature infants. This significant contribution warrants publication, albeit with a few essential modifications.

1. It is crucial that all abbreviations throughout the manuscript are accompanied by their full names upon their first usage. This step will enhance the clarity and accessibility of the content for readers.

2. The manuscript would benefit from a thorough review of punctuation to ensure adherence to English punctuation conventions. Proper punctuation usage is essential to maintain the overall readability and professionalism of the document.

3. The authors should refine the statement of the study's purpose to accurately reflect the study's core focus and objectives. A clearer and more precise articulation of the research aims will provide readers with a better understanding of the study's scope.

4. The manuscript's overall English language quality requires improvement. To address this, it is highly recommended that the authors seek assistance from a native English speaker or engage a professional language editing service. Enhancing the linguistic quality will not only improve comprehension but also elevate the manuscript's scientific rigor and presentation.

Experimental design

Null

Validity of the findings

Null

Additional comments

Null

---

## Round 0.2 · accepted · Accept

It is a good revision, I think.

Reviewer 1 ·

Basic reporting

The article must be written in English and must use clear, unambiguous, technically correct text. The article must conform to professional standards of courtesy and expression.

Experimental design

The submission should clearly define the research question, which must be relevant and meaningful. The knowledge gap being investigated should be identified, and statements should be made as to how the study contributes to filling that gap.

Validity of the findings

The data on which the conclusions are based must be provided or made available in an acceptable discipline-specific repository. The data should be robust, statistically sound, and controlled.

Additional comments

none

Reviewer 2 ·

Basic reporting

none

Experimental design

none

Validity of the findings

none

Reviewer 3 ·

Basic reporting

no comments

Experimental design

no comments

Validity of the findings

no comments

Additional comments

no comments